# Earth’s Complexity Is Non-Computable: The Limits of Scaling Laws, Nonlinearity and Chaos

**DOI:** 10.3390/e23070915

**Published:** 2021-07-19

**Authors:** Sergio Rubin, Michel Crucifix

**Affiliations:** Georges Lemaître Centre for Earth and Climate Research, Earth and Life Institute, Université Catholique de Louvain, B-1348 Louvain, Belgium; michel.crucifix@uclouvain.be

**Keywords:** Earth’s climate system, global circulation models, complex system, autopoiesis, (M,R)-system, closure to efficient causation, self-reference

## Abstract

Current physics commonly qualifies the Earth system as ‘complex’ because it includes numerous different processes operating over a large range of spatial scales, often modelled as exhibiting non-linear chaotic response dynamics and power scaling laws. This characterization is based on the fundamental assumption that the Earth’s complexity could, in principle, be modeled by (surrogated by) a numerical algorithm if enough computing power were granted. Yet, similar numerical algorithms also surrogate different systems having the same processes and dynamics, such as Mars or Jupiter, although being qualitatively different from the Earth system. Here, we argue that understanding the Earth as a complex system requires a consideration of the Gaia hypothesis: the Earth is a complex system because it instantiates life—and therefore an autopoietic, metabolic-repair (M,R) organization—at a planetary scale. This implies that the Earth’s complexity has formal equivalence to a self-referential system that inherently is non-algorithmic and, therefore, cannot be surrogated and simulated in a Turing machine. We discuss the consequences of this, with reference to in-silico climate models, tipping points, planetary boundaries, and planetary feedback loops as units of adaptive evolution and selection.

## 1. Introduction

It is generally agreed that the Earth system is complex and that this complex character must be appreciated when modelling it and taking decisions that may influence its evolution.

In present-day physics that which is called a ‘complex system’ is often the one that exhibits a large range of spatial scales, and often modelled as exhibiting non-linear chaotic response dynamics, turbulence, and power-scaling laws. It is also often assumed that this ‘complex’ character emerges from the non-linear interactions between the system’s components, and that adding more components into the system may gradually increase its complexity. At the core of this idea of complexity is the Von Neumann argument that there are degrees of complexity and that the transition from less to more complex dynamics is essentially a matter of degree of nonlinearity, connectivity, and size: “*there exists a critical size below which the process of synthesis is degenerative, but above which the phenomenon of synthesis, if properly arranged, can become explosive (complex)*…” [1] (p. 66).

In relational biology and the biology of cognition, however, the complexity is not a matter of degree of nonlinearity, connectivity, and size. As the mathematical biologist Robert Rosen argued, complexity “*has nothing to do with more complication, or with counting of parts or interactions; such notions, being themselves predicative, are beside the point...Just as ‘infinite’ is not just ‘big finite,’ impredicativities are not just big (complicated) predicativities. In both cases, there is no threshold to cross, in terms of how many repetitions of a rote operation such as ‘add one’ are required to carry one from one realm to the other, nor yet back again*” [2] (p. 124). That is, the complexity of natural systems depends whether they have or involve ‘impredicativities’ in their causality or causal organization rather than on whether they have a higher or lower dynamical order, such properties, being themselves predicative. In mathematics or formal systems, something that is impredicative (in casual terms: it knows itself) is a self-referencing definition or self-referring formal system, i.e., systems whose definitions in set theory would have to invoke what is being defined, or other things that contain the thing being defined. It turns out that self-referential systems are essentially non-syntactic (non-algorithmic) and, therefore, cannot be implemented (simulated), even in an advanced Turing machine [3].

In science and, in general, in biology, self-referentiality is expressed in the living organization as the cause and effect of itself, or more specifically, as self-production by closure to efficient causation. An efficient cause is that which constrains or drives changes. The closure to efficient causation is key to modeling the self-referential causal organization of the living systems, which is embodied in autopoiesis (auto = self, poiesis = production) and is modeled by the (M,R)-system (M = metabolism, R = repair). These are complementary explanations of the *same* self-referential causal organization of the living systems (see the next section). It turns out that autopoiesis and the (M,R)-system are self-referential systems and therefore are non-syntactic (non-algorithmic) as well.

In this article, we will argue that the Earth system qualifies as a complex system because it instantiates autopoietic organization and therefore the closure to efficient causation at the planetary scale. That is, the Earth systems, as a physical system, realize the formal pattern of an impredicative system. Some consequences of this approach are discussed in reference to in-silico climate models, tipping points, planetary boundaries, resilience, and the notion that adaptive evolution and selection operates on non-reproducing self-perpetuating planetary feedback loops.

## 2. The Road to Complexity: The Protein Folding Paradox and Living Organization

The autopoietic characterization of living systems organization is based on processes of molecular production that result in the constitution of the system itself…“*a network of processes of production…of components which…through their interactions and transformations continuously regenerate and realize the network of processes (relations) that produced them*” [4] (p. 78). The basic idea is that a molecular metabolic network generates its own semi-permeable boundary; and hence, its limit, which maintains the metabolic network occurring inside far from dissipation (Figure 1a). Both the metabolic network and the semi-permeable boundary are interdependent on each other, as part of the same self-production process. This self-production is the realization of a self-referential system in the sense that it... “*involves an iteration of the very process that generates it […] with enough entailment to close the realization process up on itself*” [2] (p. 203).

While in autopoiesis this causal organization is described as self-production by operational closure, as an (M,R)-system it is modeled as self-fabrication by closure to efficient causation. It turns out that this causality of biological systems implies a non-syntactic (non-algorithmic) character, and thus one that cannot be implemented (i.e., simulated) in a Turing machine [2,5]. In addition, it constitutively involves cognition, autonomy, and anticipation [4,6,7]. 

One of the most concrete examples of living systems being complex because they are non-computable is the protein-folding problem. The three-dimensional form of the protein shape (its molecular phenotype) cannot be obtained from the ‘information’ in the DNA sequence. Even knowing the translated polypeptide sequence and the potential physicochemical landscapes of the folding configuration, the right phenotype of a folded protein given by experimental crystallization cannot be computed, and hence simulated, in silico. This is because the folding protein problem is, indeed, impredicative or ‘paradoxical’ in the syntactic, predicative world of the computable approach to complexity. However, if we frame it in terms of *self-production* by *closure to efficient causation,* the folding protein problem disappears in the self-referentiality of the biological causal organization (Figure 1b). 

Therefore, the only way to obtain the three-dimensional form of the protein shape is essentially non-syntactic (non-algorithmic) in character. In what follows, we will argue that both the autopoietic and the (M,R)-system are sufficiently general to provide a possible road to showing how the Earth qualifies as a complex system and what this implies for the future evolution of the system.

## 3. The Autopoietic Complexity of the Earth System Organization

We consider that the Earth’s complexity resides in its causal organization, which is self-referential. The road to demonstrating that the Earth qualifies as a complex system thus passes through a consideration of the Gaia hypothesis: that the Earth is an instance of life and therefore an instance of biological organization at a planetary scale. 

Numerous authors agree that autopoiesis is a plausible scenario for the instantiation of life organization on a planetary scale [8,9,10,11,12,13,14,15,16,17]. Recent work showed that this is plausible in a formal syntactic framework, using a proof-of-concept based on the chemical organization theory and the zero deficiency theorem applied on a simple but representative Earth molecular reaction network [18]. These results show that the Earth is an organized system, and this organization may approximate to an autopoietic system, making the Gaia hypothesis tractable from this standpoint. 

An intuitive but reasonable road to elucidate the Earth’s organizational system and whether this organization is autopoietic is derived from two rationales: At whatever scale, the physical embodiment of autopoiesis, either in the cellular, metacellular, or in this case, in the planetary domain, must always be molecular: “*There are autopoietic systems of higher order (metacellulars or Gaia), integrated by* (*populated by*) *lower order autopoietic unities that may not be the components realizing them as autopoietic systems... there are higher order autopoietic systems whose components are molecular entities produced through the autopoiesis of lower autopoietic unities*” [19] (p. 53, brackets and underline are mine). This is also indicated elsewhere [20,21].Autopoiesis must involve an interdependence between a metabolic network and a semi-permeable boundary. On the planetary scale, this can be interpreted as the biosphere (involving the lithosphere and hydrosphere) being the metabolic network, and the atmosphere being the semi-permeable boundary, respectively.

The early systematization of observations already shows that most, if not all, the atmospheric molecular components of the troposphere and stratosphere, key atmospheric layers for climate dynamics, are metabolically produced [22,23]. Morowitz points out a self-evident truth: “*all organisms interact through the gas-phase components that they take up from and give off to the atmosphere*…” [24] (p. 5). That is, the biosphere depends across time on the atmospheric composition, upon which it metabolically operates. This metabolic production involves the lithosphere and hydrosphere [25]. Therefore, throughout the history of the Earth there has been a molecular interdependence between the biosphere and atmosphere, and this is self-referential in the sense that the continuous metabolic fabrication of atmospheric molecular components takes place in the same domain which allows metabolism to continuously operate (Figure 2a). The mutual specification of the atmosphere and the metabolic reaction network offer an explanatory account of a self-producing organization that amounts to a self-referential system on the planetary domain, thus a form of planetary autopoiesis. This is also reflected in the co-definition (co-specification and co-production) of the cell–Earth system.

The Earth’s complexity as such is precisely reflected in the challenge posed by the projects of planetary terraforming. It is considered that planets such as Venus or Mars could be bombarded in order to produce an Earth-like atmosphere. However, there is no guarantee that any biosphere could thrive there and maintain such an atmosphere. On the other hand, bringing to Mars biosphere-like stations, such as in Biosphere 2, in order to produce a ‘habitable’ atmosphere within them is considered. Yet, these stations are, for some extent of time, closed-sustainable systems and there is no clear idea how to open up them to connect with the non-habitable Martian atmosphere, in order to accomplish the situation described by Morowitz above. There is an idea to ‘seed’ the exo-planets with extremophiles microbes and let them to ‘colonize’ it. However, the autopoietic organization of the Earth system and experiments with extremophiles in space strongly suggest that living systems need not just an environment but a bioenvironment in order to persist, which involves a form of organism–bioenvironment self-reference (Figure 2b). This is what is meant by the “*organism-niche unity*” [26]. 

The fact, at least mathematically, is that there is no passage from systems that are not self-referential to systems that are. That is to say, it is a causal impossibility to generate a system that in principle is closed to efficient causation (operational closure), and therefore complex, from algorithmic (non-self-referential) systems. With that said, now we will turn to providing further opportunities to show how the Earth’s complexity as an instantiation of autopoiesis at the planetary scale can be compared with in-silico climate simulations, as if the Earth was modeled by (surrogated by) what in Robert Rosen’s mathematical biology is called the (M,R)-system.

## 4. The (M,R)-System and the In-Silico Earth’s Climates of Global Circulation ‘Models’

The notion of complexity is much used in Earth climate science. Differently from what we have exposed so far, that is, that the complexity resides in the system self-referentiality or impredicativity, in present-day physics, it is generally considered that it even resides in deceptively simple (but non-linear) systems, which may exhibit sensitivity dependence to initial conditions. In this case, the tiniest uncertainty in initial conditions propagates in time so that predictions are limited to a narrow window of time, because predictive capacity decays exponentially. However, large non-linear dynamical systems are expected to be chaotic, which also produces unpredictability. Paradoxically, though, a closer look at that which present-day physics considers ‘complexity’, but in reality is only algorithmic and therefore only complicated, comes to the rescue. 

Complicated systems have been proven to display structures at different spatial-temporal scales. It is therefore generally possible to describe the evolution of macro-structures without knowing exactly the state trajectory of the system’s state at the smallest scale. This argument has been well described, and formalized mathematically, based on the time-scale separation assumption, and it provides formal support for the idea that it is reasonable to attempt to predict, for example, the next glacial inception, even though mid-latitude weather cannot be predicted precisely beyond two weeks (as demonstrated by any weather forecast).

There is however nothing in this description that challenges the idea that the dynamics of the system can, at least in principle, be deduced from the laws of mechanics at the smallest scale, hence from algorithmic programs. This important assumption justifies the character of programs of research and prediction using general circulation models (GCM). A state-of-the-art GCM is a dynamical system with a state vector of well over 10^6^ variables, and the rules for the transition of these state vectors from one-time step to the next are encoded in algorithmic programs that include hundreds of thousands of lines of code (see the IPCC reports). Thus, GCMs are, in essence, very large systems of time-difference equations that are translated as algorithms and executed by ‘supercomputers’. That is, it is an in-silico climate simulation. Standard GCMs produce models of the form *F*(*A*) = *B*, and this approach has been successful in introducing several important concepts. 

A non-linear dynamical system may have sensitive dependence on initial conditions. The *B* is then a “strange attractor”, which indeed has a non-trivial topology.It may also happen that small changes in the system parameters (included in *A*) result in changes in the topology (the “shape”) of *B*. This is a bifurcation.In turbulence theory, one exploits symmetries in the equations governing mechanical flows to deduce that *B* should have properties of scale invariance.In the broader setting of statistical mechanics, one seeks quantities that are conserved globally (energy, entropy), applying principles of statistical inference (typically, the maximum entropy principle) to deduce distributions. Hence, *B*, the output, takes the form of distribution functions.In a GCM, the computing of *F* on a supercomputer takes long (it may take months), and the output *B* is stored in mass storage facilities of terabytes of data. Climate observers need time to analyze them, identify “mechanisms” (like sea-ice feedback), and discuss them.

Although such a program of climate simulation is well-established, there is a concern about the conclusion that the principal limit to the accuracy of the description of the Earth’s complexity is resolution; hence, computing power. That is, it is assumed that the Earth’s complexity could, in principle, be surrogated by a numerical algorithm if enough computing power were granted. However, as we have shown, the Earth’s complexity resides rather in its autopoietic (self-referential) organization, and therefore the Earth may escape algorithmic representations. Let me clarify this further. 

The (M,R)-system is a formal model and theory to capture the self-producing causal organization of biological systems based on the mathematical language of category theory [27]. The (M,R)-system has been shown to generalize the causality behind the autopoietic organization of living systems. However, the (M,R)-system theory is very different to that used in GCMs. Thus, to compare in-silico climate simulations with the Earth’s complexity, as surrogated by the (M,R)-system, we will introduce some basic concepts and notations by reference to the GCM iteration *x_i_*_+1_ = *F*(*x_i_*). To make the discussion a bit smoother, we will drop the reference to the parameter *ψ*, and consider it as part of the specification of *F*. When we mention a modification of *F*, we mean either a modification of the equations of the simulation, or of its parameters. 

When we claim that we are ‘*modeling*’ climate dynamics, we claim that *x_i_*, *x_i_*_+1_, and *F*, which are defined as mathematical objects, have their counterparts in the climate system. This means that at least some components of *x_i_* can be *observed* (perhaps indirectly, via an observation operator). We also consider that there is a relationship between what can be observed at time *t_i_*, and at time *t_i_*_+1_, and that this relationship can be computed with the algorithm *F*. We can rephrase this by stating that if the space of possible states for *x_i_* is *X_i_*, then *F* defines a range of possible states for *x_i_*, and this range can be noted *X_i_*_+1_. The standard notation is *F*:*X_i_*
→ *X_i_*_+1_. However, in the (M,R)-system model, Rosen used a non-standard notation: *F*
→ *X_i_*
⇢ *X_i_*_+1_. The notation is a proxy for the efficient (→) and material (⇢) causes, which allows us to clarifying the system’s causal categories.

In the biological context surrounding the development of the (M,R)-system model, *F* is identified as a material efficient cause (it can represent the active site of an enzyme; however, the principle is general enough to apply to other physical instantiations) and in a given environment, constrains a material transformation such that it selects elements of the environment *A* and transforms them into *B*. Following the above notation, this reads *F*
→ *A*
⇢ *B*. In in-silico computation, for example, the function *F* is coded in memory as a suite of binary states which, in the *syntactic* of the programming language (which provides a context), generates the mapping of *X_i_* onto *X_i_*_+1_. This describes what materially happens when the numerical simulator is run. At this point, the object *F* may be seen either as a material structure or as a function. 

The point of the ‘R’ in the (M,R)-system is that the metabolism *F* is undergoing wear and tear, and therefore needs to be repaired. The organism does this, and the (M,R)-system models this as a repair function, which is symbolized by introducing a new repair function *φ*. This function takes *B* as a material cause to produce the *F*, notated as *φ*
→
*B*
⇢
*F*. In a standard GCM this might appear as an incongruity: *F* is first seen as a “function” or efficient cause, which produces the transformation *A*
⇢
*B*, but next it is seen as the material cause or initial product of a transformation of *B*. GCMs are not designed to support this double entailment, but there are mathematical formalisms, like lambda calculus, that would support it. In functional programming a function can also be the output of a function. However, the components of the (M,R)-system, such as *F* for example, serve not just as the efficient cause of *B*, but also as the material and the result (final cause, output, or anticipation) of *φ*, which does not have a parallel in any syntactic functional programming of in-silico simulations. For example, an enzyme or an organ can simultaneously be seen as a constraint (efficient cause), as a component that needs to be repaired or replaced (material cause), and as a functional element (result). 

Usually, when a version of a GCM or any dynamical representation of the climate system is released, *F* is frozen. This is so, because in the specification of *F*, the processes that may affect climate change in the future are ignored. Therefore, to some extent, *F* may become unsuitable or unacceptably inaccurate at some stage, and processes that have been overlooked may appear to become important in the future. Computing *B* (which is the model “output”) does not pose any specific mathematical difficulty when *F* is specified as an algorithm and frozen. The one who specifies an *F* is the climate modeler(s). The output *B* depends on (reasonable) adjustments to *F* to bring *B* into closer alignment with a desired target (e.g., hypothetical climatic-change scenarios). These adjustments are classically justified as part of the quality control process (bug tracking and fixing) and model tuning. Yet, the climate modeler(s) can implement a procedure of automatic tuning of *F*, designed such that *B* matches observations about the climate system. In that case, we could say that *φ* is a *computer code* that implements an algorithmic process that implements automatic tuning. However, *φ* still needs to be produced/defined (and corrected), perhaps by a statistician or a numerical analyst, an external agent at the end. We could keep iterating in this way, without changing the nature of the conclusion: at some point we need someone external to the system who specifies *F*. 

The situation is different in the (M,R)-system model, which entails closure to efficient causation of *F.* The metabolism function *F* and the repair function *φ* are generated from the inside, rather than specified from the outside. To make this argument concrete, the autopoietic organization of living systems is self-referential. In other words, a subset of the organism has to play the role of *β*. It generates *φ* using internal information (or anticipative model) of what *B* should be, using a subset of the metabolism *F*. Stated mathematically, *β* is a function that satisfies the definition *β*: (*B*, *F*) → *φ*. It is possible to review this definition by invoking a formal mathematical act called “currying”, common in functional programming: *β* is redefined as a function of *B*, which generates a function of *F*, the output of which is *φ*. That is: *β*: *B* → *F* → *φ*. Equivalently, as *β*(*B*) is a function, we can write: *β*(*B*): *F* → *φ*. Now using the Rosen notation, this gives *β*(*B*) → *F*
⇢ *φ*. This notation carries the meaning that *B* provides the structural information (such as active sites of enzymes) for the production of *φ*. This is represented in the (M,R)-system by a synthesized relational mapping with a *closed* directed graph that uses the two categories of entailment defined by → and ⇢ (Figure 3). 

On the (M,R)-system graph, *B* and *β*(*B*) occupy the same node, and this assumes implicitly that *β* is a well-defined function. *B* is an output (final cause) of *F*, and is also a material cause of *F* and efficient cause of *φ*. Such a concatenation of entailments generates a global stability that may be thought of as if *B* were a goal function. It is in this sense that the system has acquired a quality of “autopoietic unity” that distinguishes it from systems that do not achieve closure to efficient causation, such as GCMs. 

Observe that if *β* is strictly surjective, the destination of the inverse function of *β*(*B*), denoted *b*, is a strict subset of *B*. Only a subset *B* may be involved in the production of *φ*. In this case, the subset *b* is sufficient (contains enough information) to imply *F*. The conclusion is that *b* implies a set that contains itself. Put in colloquial language, a subset of *B* needs to be “aware of” (informative about) the whole of *B* and how it is being produced. This is where impredicativity comes in. We can see it at work: *B* depends on *F*, which is constantly being replaced, by processes which are critically dependent on *φ*. Yet, as we just noted, *φ* depends on only a subset of *B*. 

The dynamical realization of the (M,R)-system model cannot be implemented in the algorithmic language of dynamical systems [28]. Thus, in general we reach the same conclusion when analyzing a GCM under the prism of the (M,R)-system. If the *F*, for example, is seen as the specification of a dynamical system, then *B* would be its time-invariant measure (roughly said, the attractor) and *A* would represent an external forcing. Specifying what *φ* is less straightforward: we need a mapping *φ* of attractor measures onto some coded specification of *F*, but also a function *β*(*B*) that will produce *φ*. At this point the dynamical system turns out to be open to efficient causation. 

In this sense the (M,R)-system model provides us with further opportunities to formalize the proposal that the complex condition of the Earth system can only be explained by the instantiation of closure to efficient causation at the planetary domain. This expression of the Gaia hypothesis, although tentative, gives us the opportunity to illustrate some of the theoretical aspects underlined above. A potential (M,R)-system model of the Earth system would have to be consistent with the observation that whatever *B* can be in the Earth system, it is itself involved in a chain of entailments that cause *F*, which is out of the scope of GCM models of the form *F*(*A*) = *B.*


The key to Rosen’s views on complexity is that the properties of a natural system are subsequently discussed in terms of the models that a natural system can have. Consequently, complex systems are ones that have complex models and the (M,R)-system is postulated to be one of them [29,30,31]. Thus, we can ask whether the Earth system is implementing the (M,R)-system, and therefore qualifies as a complex system; a question which is not addressed by the current appraisals of Earth-system complexity.

Turbulence, nonlinearity and chaos are often seen as synonymous with Earth’s complexity; however, by definition they are mathematical images that are implementable in a Turing machine, and are therefore simulable. Beyond technical controversies, impredicativity in the (M,R)-system cannot be dealt with by classical methods of programming [32]. This implies that one cannot (easily) provide an iteration which satisfies the causal entailments of the (M,R)-system. Conversely, the (M,R)-system is richer in entailments (causation), to the extent that it cannot be implemented in a Turing machine, is non-simulable, and therefore is complex [5,33]. The proposal opens the possibility of an entirely new research program to understand the Earth’s complexity in terms of organization, allowing us to understand the fundamental difference between what we should call Earth’s *complexity*, and the situation of *complication* described by GCMs. Having suggested this, in the following section, we will discuss the properties and consequences of the Earth’s complexity in terms of the autopoietic, (M,R)-system organization, and what this implies for tipping points, planetary boundaries and resilience, and for the proposal that adaptive evolution and selection operates on non-reproducing self-perpetuating planetary feedback loops. 

## 5. Properties and Consequences

### 5.1. Earth Complexity Is All or Nothing

The analysis above implies that biological organization is all or nothing. Autopoiesis happens or not [19]; thus complexity either occurs or it does not, and there are not intermediate degrees of complexity. The system realizes self-production or the system falls apart. However, the molecular embodiment of autopoiesis does not mean that organization may be reduced altogether to the molecular phenomena of chemical reaction networks. Rather, it simply points out a fundamental distinction about what should be the Earth’s complexity in reference to the question of size, degree, and connectivity compared with impredicative systems. It is in this precise sense that the realization of the autopoietic organization on a planetary domain could allow us to claim that the Earth system needs to be qualified as complex and not merely complicated, i.e., having neither increasing degrees of, nor more or less, complexity. 

### 5.2. Earth Complexity Implies Conservation of Organizational across Structural ‘Tipping’ Changes

One of the implications for understanding the Earth’s complexity from the characterization of autopoietic systems is the difference between structure and organization of the system [4,34]. This difference between structure and organization is related, in formal terms, to the fact that biological systems are open to material cause, yet closed to efficient causation. The first relates to the dynamic and thermodynamic, while the second relates to the organization of biological systems.

Structural change and organizational conservation are the keys to complex system dynamics and modeling. The structure may undergo changes as long as the autopoietic organization is preserved [19,26,34]. Different structures (scenarios of Earth’s history) correspond to the structural change, but with conservation of the Earth’s self-producing organization. Structural change is closely associated with stability and it is usually assumed to be a general property of dynamical systems. Some authors understand the stability of the Earth system as self-organization, alternate states (multistability), thresholds, and early signals of change [35,36,37,38]. One might conjecture that under this understanding, the so-called “tipping points” [39], including the potential cascades, may be regarded as structural changes that so far have not caused the loss of the Earth’s organization. In other words, the Earth system can go through different structural changes (extreme, abrupt, catastrophic) while preserving its autopoietic organization, and hence its complexity; these “tipping points” can be, so to speak, non-fatal. Indeed, Earth’s history has been punctuated by several “catastrophic” structural changes, such as the transition from reductive to oxidative atmosphere [40], the mass extinction (diminished biosphere) of 50 to 90% of diversity [41], planetesimal impacts, and geomorphological changes. Yet, the complex (living) character of the Earth system has persisted. 

### 5.3. Earth Complexity Implies Multiple Structural Relations Carried out by Multiple Components

Many of the components of an (M,R)-system serve as outputs (final cause), as efficient causes, and also as material causes of other components. Structural changes of the Earth system understood as an autopoietic system may have many structural interdependencies. If one structural dimension in the Earth system is changed (e.g., a tipping point of Greenland ice melting), the complete system may undergo correlative changes in many structural dimensions (e.g., possible tipping cascades). Such structural changes can suppress, allow, or create new components and relations (processes and constraints) [19,26,34]. Therefore, these components may be integrated into the system with different structural relations, either as processes or as components realizing the constraints on the processes. That is, components may have multiple structural relations, and structural relations may be carried by multiple components. For example, ice-sheets have multiple structural relations with different processes and components of the Earth system, linked as they are to climate dynamics, the nutrient cycle, the ocean crust, and the water cycle [42,43,44,45,46].

Nevertheless, there may be structural changes that could make the Earth system lose its organization and thus enter into an ‘*autopoietic oscillator death*’ [47]. This touches on the definition of what could be a ‘critical’ perturbation for the complexity of Earth’s organization that may break down its autopoiesis and thus be fatal, and whether the thresholds of the ‘planetary boundaries’ [48] are critical for the planetary self-producing organization. 

### 5.4. Earth Complexity Is More Than Input–Output Control Feedback Systems

Recent proposals suggested that adaptive evolution and selection operates through non-reproducing self-perpetuating planetary feedback loops [49,50] and that in general feedback loops are key for climate dynamics, and hence Earth complexity. However, mechanisms of feedback self-regulation have been described on so-far lifeless planets, such as Mars [51,52], and seen as stabilizing the surface temperature of a lifeless Earth [53]. 

Feedback loops are a legacy of the first order cybernetics and control theory [54,55,56] that was developed as a mathematical framework for what Walter Cannon named the *error-correcting theory* of regulation or ‘*homeostasis*’; stability through constancy [57]. The underlying theorem is that ‘every good regulator of a system must be a model of that system’ [58]. The core idea is that there is an input-output system that reaches *stability through self-regulation by negative and positive feedback loops* [54,55,56]. The feedback loop is an error-counteracting response, which takes place *only* when there is external perturbation (forcing/input) sufficient to make the system’s parameters deviate from pre-defined ‘set points’. It is said that cybernetic systems behave as goal oriented systems, because they return to their ‘set points’ once they are perturbed. That is, the error-counteracting response is a reactive response. Moreover, the relation of input to output implies that external forcing determines what happens inside a system, such that a forced system will generally end up tracking the forcing. The explicit relation between the two is embodied in the engineering transfer function of the system. That is already suggestive, but it is very risky to simply extrapolate such ideas of a simple or even a complicated system, that when a system is fabricated by closure to efficient causation, i.e., it is complex, because the Earth system as such may entail the absence of input and output controls, and remain organized through autonomy and anticipation. 

### 5.5. Earth Complexity Involves Autonomy and Anticipation

The autopoietic system is wide open to imposed forces in a time-independent manner and has a structure that changes following a course contingent on the course of its interactions. However, forcings that may impinge upon the system may trigger structural changes without specifying them [19,26,34]. Even if a forcing causes continuous structural changes in the system, the specific nature of these changes may be determined not by the forcing (input), but rather by the autonomy of the autopoietic system [59]. Indeed, an autopoietic system may build up predictive or anticipatory models of the forcing in order to act autonomously and predictively over such forcings. 

Autonomy is a property of systems upon which the flows from environment to system, and from system to environment, are determined by what is inside the system [2,59], and such that everything that happens in the system or to it is determined in it at every instant by its structural dynamics at that instant [19,26,34,59]. Anticipation is the behavior of avoiding a predicted deviation, which is energetically much cheaper than correcting a deviation (feedback), whether through fluctuation-counteracting or fluctuation-amplifying. This implies that feedback responses are reactive and cost-ineffective responses. Anticipation is based on internal predictive models that living systems make of their environment and themselves, throughout their ontogeny and phylogeny [60,61], which allows changed behaivour at an instant in accord with the model’s prediction and pertaining to a future (later) instant [62]. These models involves feedforward loops and are inherent to the causal entailment organization of self-production by closure to efficient causation [62]. 

Given that autonomy is a corollary of autopoiesis [59] and that every autopoietic system minimizes free energy (active inference) [47], it is plausible that the Earth’s complexity involves autonomy and anticipation as well [63,64].

### 5.6. Managing Complex Systems Requires Complex Models

It is important to ask whether dynamical systems or GCMs ‘model’ the complexity of the Earth system to the extent that the potential intervention (the application of geoengineering) is reliable. That is, one must ask how well the mathematical machinery of non-linear chaotic dynamics, turbulence, power scaling laws, and feedbacks can inform us about potential chain disruptions and domino effects in the Earth’s organization if geoengineering is applied. 

Based on the arguments about the properties and consequences of complex systems presented here, it is expected that under a geoengineering perturbation (solar radiation management or carbon sink), the Earth system may undergo correlative changes in many structural dimensions. Thus, different Earth components will rearrange in different multiple structural relations in order that the Earth’s organization remains autopoietic. These rearrangements will be in general a kind of self-structuration, since any Earth response to any perturbation, including geoengineering, is autonomous, i.e., it goes beyond input–output feedback loops [18]. Moreover, it is highly plausible that the Earth system will track such geoengineering, forcing and building up anticipative models of it, and thus there will be no clear idea about how this self-structuration would take place. It is therefore extremely important to take into account complex models such as the (M,R)-system to model the Earth’s system rather than only simulating it.

## 6. Conclusions

We understand here that the complexity of the Earth lies in its *biological organization* rather than in its manifestation of power scaling laws, nonlinearity, and chaos. The present terrestrial environment is itself the cause and result of its own fabrication processes, with no separation, at geological scales, between product and producer, between biotic and abiotic elements.

This implies that the Earth, when understood as a complex autopoietic, anticipatory system, features a horizon of indeterminacy that must be distinguished from the horizon of predictability commonly attributed to algorithmic programs of dynamical systems. This program may be limited or just a shorthand approximation of the Earth’s complexity. This may be consistent with the assertion that there exists no equivalent to thermodynamic constraints and feedbacks mechanisms by which we can predict the anticipatory autonomous behaviors of the Earth system. This understanding of Earth complexity may represent, thus, a step forward from current programs, based as they are on the reactive paradigm of feedbacks, dynamical systems, and algorithms in general.

It turns out that Earth complexity embodies a unique attempt to prove that the closure of metabolic networks at the planetary scale must satisfy certain regularities of organization that go beyond reactive, ‘complicated’ models. These regularities, arising from Earth complexity, as summarized in the properties listed above, suggest an effective system fabrication that generates self-referential mathematical objects. In other words, the relation between Earth complexity and power scaling laws, feedbacks, nonlinearity, and chaos may be compared to the situation faced by early cartographers, who were attempting to map the surface of a sphere while armed only with pieces of (tangent) planes. “As long as they only mapped local regions, the planar approximations sufficed; but as they tried to map larger and larger regions, the discrepancy between the map and the surface grew as well. If they wanted to make accurate maps of large regions of the sphere, they had to keep shifting their tangent planes. The surface of the sphere is in some sense a limit of its planar approximations, but to specify it in this way requires a new global concept (the topology of the sphere; i.e., its curvature) that cannot be inferred from local planar maps alone” (Rosen 1985). It turns out that complicated algorithmic simulations are the planar approximations, and the Earth’s complexity is in some sense a limit of its planar approximations, which leads us to widen our concept of what Earth’s complexity is, or should be.

## Figures and Tables

**Figure 1 entropy-23-00915-f001:**
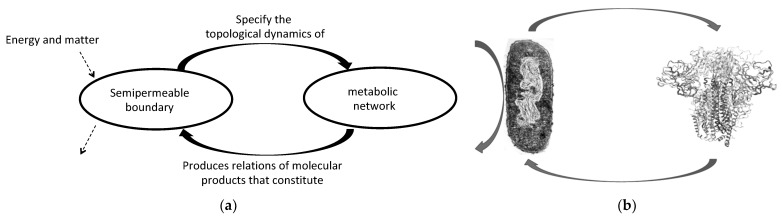
(**a**) The autopoietic system is an open system in the sense that it selects which matter and energy enter and exit the system (dotted right arrows). The operational closure occurs when a molecular reaction network produces a semipermeable boundary and this specifies the topological dynamics (molecular concatenation) of the metabolic reaction network (arrows in both directions); that is, when there is an interrelation between the metabolic network and the boundary; (**b**) The protein folding only takes place as a self-referential relation of cell–protein, such that cells cannot operate with defective protein shapes and the right folding only takes place within cells. Notice that the interaction with environmental energy and matter does not determine the protein folding either.

**Figure 2 entropy-23-00915-f002:**
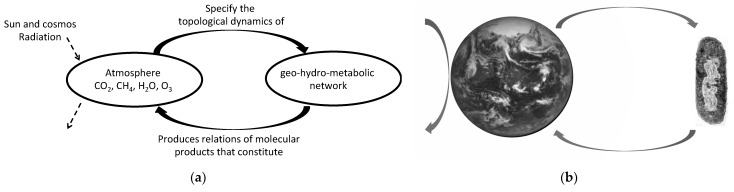
(**a**) Planetary-scale autopoietic organization takes place in the interdependence between the geo-hydro-metabolic network and the atmosphere (Earth’s semipermeable boundary), in the sense that the geo-hydro-metabolic reaction network produces the main components of the atmosphere, and the atmosphere specifies and allows the dynamics of the geo-hydro-biospheric metabolic network (solid horizontal arrows). The Earth system, as an open system, exchanges matter and energy with its space environment (solid vertical arrow in the left of the earth); (**b**) the organism (cell) and the Earth (bioenvironment) coupling is also an impredicative system, similar to the cell–protein folded self-reference, in the sense that both exists thanks to each other.

**Figure 3 entropy-23-00915-f003:**
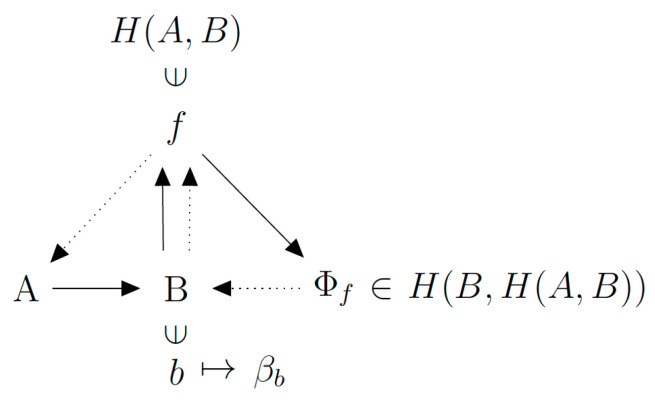
The mathematical structural organization of the (M,R)-system.

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
