# Peer review of "Earth’s Complexity Is Non-Computable: The Limits of Scaling Laws, Nonlinearity and Chaos"

_entropy, 2021, doi:10.3390/e23070915_

Round 1
Reviewer 1 Report
Overview:This is a perspective paper that offers a philosophical view of the Earth system andhow it should be modeled. I was not able to understand the main point of the paper well enough togive a decent summary. I recommend the following in order to make the paper more clear:
- Try to avoid jargon and explain things in as simple terms as possible.
- Work with a native English speaker to improve the grammar and writing.
- Fix the figures. In the pdf I got they are rendered incorrectly.
Since the paper has been submitted to a scientific journal, I recommend making some scientificpoints in addition to the philosophical ones. Here are some practical ways to do this:
- Make some sort of calculation to demonstrate what you are talking about and what it couldbe useful for. This could be with a low-order model, or with a more complex one. It couldalso be for a toy system.
- Make some falsifiable predictions that could be tested.
I am not recommending that the paper be rejected because it’s possible that it is making asubtle and intricate point that I don’t have the training to understand; however, it does need majorrevisions.
Author Response
Overview: This is a perspective paper that offers a philosophical view of the Earth system and how it should be modeled. I was not able to understand the main point of the paper well enough to give a decent summary. I recommend the following in order to make the paper more clear:
R-Thank you very much for pointing out that the manuscript, in its previous state, was not comprehensible to the public who do not necessarily know the impact of the Godel's incompleteness theorems and their link to the impredicative (self-referential) systems on the definition of complexity used on the paper. In the new version I have made explicit this in the introduction.
- Try to avoid jargon and explain things in as simple terms as possible.
R-The new version of the manuscript avoids the use of jargons as much as possible, yet, I am afraid, that there are terminologies that cannot be named otherwise.
- Work with a native English speaker to improve the grammar and writing.
R-It has been edited and improvised grammatically by a native English speaker and the figures have been fixed.
- Fix the figures. In the pdf I got they are rendered incorrectly.
R-As you are the only reviewer who has not been able to observe the figure well, if this happens again, it is very possible that it is a problem with acrobat/word versions. If this is the case, I can provide the pdf version directly, to avoid such problem.
Since the paper has been submitted to a scientific journal, I recommend making some scientific points in addition to the philosophical ones. Here are some practical ways to do this:
- Make some sort of calculation to demonstrate what you are talking about and what it could be useful for. This could be with a low-order model, or with a more complex one. It could also be for a toy system.
R-The paper presented here does not offer a philosophical view of the Earth’s complexity. Rather it is concept-based paper, rather than method-based. That is, the information it carries is about why a natural system (the Earth) is complex, rather than how. I also develop the implication of why it is a complex system. It turns out that, any sort of high or low-order simulation may contradicts the main point of the manuscript, since they belong to the realm of predicative systems, which inherently are simulable by any sort of Turing machine. The paper is claiming exactly the opposite; the Earth system is complex because is an (M,R), autopoietic system, which inherently is a self-referential system which by formal definition cannot be surrogated by an algorithm (a computational simulation).
This is extensively developed in the Robert Rosen’s book Life Itself, yet, already the Godel's incompleteness theorems showed that Hilbert's program to find a complete and consistent set of axioms for all mathematics is impossible. That is, no consistent system of axioms whose theorems can be listed by an effective procedure (i.e., an algorithm, either of high or low-order) is capable of proving all truths about even the arithmetic of natural numbers. For any such consistent formal system, there will always be statements about natural numbers that are true, but that are unprovable within the system.
In other words, surrogate the Earth system to a predicative toy simulation, is the same that to formalize number theory. More syntax is done on number theory; more the identity of number theory disappears.
- Make some falsifiable predictions that could be tested.
R-This is a nice point of discussion, which is being developed in a forthcoming paper. For a brief comment I think the limits of computational simulation are confronted when one tries to apply the inferential problems of predicative to impredicative (self-referential) mathematical systems – systems whose definitions in set theory would have to invoke what is being defined, or other things that contain the thing being defined. In these cases we get what Bertrand Russell called paradoxes and vicious circles, and which Kurt Gödel examined more formally in his generic incompleteness theorems, which caused foundational problems for predicative mathematics. Even machine learning fails to find a solution to such self-referential problems.[1] This poses an essential problem in the current structure of scientific discovery. Here comes the key difference when comparing present day physics and the physics of living systems.
Karl Popper argued for falsifiability and opposed this to the concept of verifiability that Harold Morowitz considers is more appropriate when it comes to the biological, thus complex systems: “Popper stresses falsification, and Margenau stresses verification, yet these are but different aspects of the same act: That which survives as scientific reality is the structure that has been verified and never falsified”[2]
Yet everyone knows – I can confirm your existence – verify it – but not prove it's impossible for you to exist – you cannot falsify the living phenomenon (as an instantiation of self-referential systems), but only verify it.
Indeed, in biology, from the, protein-folding problem, the origin of life to the Gaia hypothesis, the problem is not of falsification but of verification. One can verify whether a natural system is alive (complex) or not, not falsify it.
I am not recommending that the paper be rejected because it’s possible that it is making a subtle and intricate point that I don’t have the training to understand; however, it does need major revisions
R-I'm sorry I wasn't explicit about making the subtle point about complexity understandable. I hope that the new version of the manuscript is more explicit in explaining what we should refer to when we talk about complexity, at least from a mathematical biology standpoint.
[1] Castelvecchi, Davide. "Machine learning leads mathematicians to unsolvable problem." Nature 565.7737 (2019): 277-278: https://www.nature.com/articles/d41586-019-00083-3.
[2] Morowitz, Harold J. Beginnings of cellular life: metabolism recapitulates biogenesis. Yale University Press, 1993.
Reviewer 2 Report
Modeling Earth system is a complex tool, as it exhibits non-linear chaotic response in dynamics, turbulence, and power scaling laws. The author studies the Earth as complex system pass through considering the Gaia hypothesis.
The paper is generally well written and the idea behind is interesting and useful for further applications. The overall scientific importance and relevance of the paper is fair, implications for further analysis and studies are not negligible.
However, before acceptation I suggest a thorough English proofreading.
Author Response
Thank you very much for your positive and very stimulating review. In the new version of the manuscript I provided sufficient background and include all relevant references in the introduction.
The manuscript has been edited and improvised grammatically by a native English speaker. I hope all these changes are sufficient for your final acceptance for the manuscript publication.